# Management of Pediatric Bilateral Vocal Fold Paralysis: A State-of-the-Art Review of Etiologies, Diagnosis, and Treatments

**DOI:** 10.3390/children11040398

**Published:** 2024-03-27

**Authors:** Jerome R. Lechien

**Affiliations:** 1Research Committee of the Young Otolaryngologists of the International Federation of Otorhinolaryngological Societies, 92150 Paris, France; jerome.lechien@umons.ac.be; 2Department of Otolaryngology-Head and Neck Surgery, CHU Saint-Pierre, B1000 Brussels, Belgium; 3Department of Otolaryngology, Elsan Hospital, 92150 Paris, France; 4Department of Otolaryngology-Head and Neck Surgery, Foch Hospital, School of Medicine, UFR Simone Veil, Université Versailles Saint-Quentin-en-Yvelines (Paris Saclay University), 92150 Paris, France; 5Division of Laryngology and Bronchoesophagology, Department of Otolaryngology Head Neck Surgery, Faculty of Medicine, EpiCURA Hospital, UMONS Research Institute for Health Sciences and Technology, University of Mons (UMons), B7000 Mons, Belgium

**Keywords:** laryngeal, larynx, otolaryngology, head–neck surgery, voice, nerve, paresis, immobility, vocal fold paralysis, reinnervation

## Abstract

Objective: This paper reviews the current literature about epidemiology, etiologies, diagnosis, and management of pediatric bilateral vocal fold paralysis (PBVFP). Methods: According to PRISMA statements, a narrative review of the current literature was conducted through the PubMed, Scopus, and Cochrane Library databases about the epidemiology, etiologies, diagnosis, and management of PBVFP. Results: PBVCP is the second most common congenital laryngeal anomaly in the pediatric population, accounting for 10% to 20% of pediatric laryngeal conditions. PBVCP is related to idiopathic (42.2%), congenital (19.7%), and neurological (16.9%) conditions. A tracheotomy is required in 60% of cases regarding stridor and dyspnea, which are the most prevalent symptoms. The diagnosis is based on the etiological features, clinical presentation, laryngoscopic findings, and objective examinations. Laryngeal electromyography may be used to support the diagnosis in difficult cases, but its reliability depends on the practitioner’s experience. The primary differential diagnosis is posterior glottis stenosis, which needs to be excluded regarding therapeutic and management differences with PBVCP. Transient surgical procedures consist of tracheotomy or laterofixation of the vocal fold. Current permanent procedures include uni- or bilateral partial arytenoidectomy, posterior transverse cordotomy, cricoid splits, and laryngeal selective reinnervation. There is no evidence of the superiority of some procedures over others. Conclusions: PBVCP is the second most common laryngeal disorder in the pediatric population. Diagnosis is based on etiological and clinical findings and may require the use of laryngeal electromyography. Therapeutic management may involve several transient or permanent surgical procedures that are associated with overall subjective improvements in symptoms, laryngeal findings, and low complication rates.

## 1. Introduction

Pediatric bilateral vocal fold paralysis (PBVCP) is a controversial issue in pediatric otolaryngology and head and neck surgery regarding diagnosis and medical and surgical management [1]. PBVCP is a rare condition with an undetermined prevalence [1,2]. The disease may affect neonates, children, or infants and may be related to a myriad of neurological, malformation, iatrogenic, or idiopathic diseases [1]. Clinically, most children present with stridor, dyspnea, hoarseness, or dysphagia, and the laryngological condition may lead to growth disorders [1]. However, in some cases, the disease may be pauci-symptomatic, which explains why many cases are not detected early [1,2,3]. Indeed, the detection commonly requires the realization of nasofibroscopic examination by an otolaryngologist or head and neck surgeon who needs to be aware of this rare condition. The management of children with BVCP is controversial for many reasons. First, the disease is often confounded with posterior glottic stenosis, which is a different condition requiring different therapeutic approaches [4]. Posterior glottic stenosis is also called cricoarytenoid joint ankylosis, where the laryngeal recurrent nerves are both functional but the mobility of the cricoatytenoid joints is impaired [4]. Consequently, some surgical approaches, such as laryngeal reinnervation, do not make sense, whereas others, such as partial antero–medial arytenoidectomy, are insufficient to improve the airway [4]. Second, most studies are retrospective uncontrolled chart reviews that do not report evidence-based data for the most effective transient or definitive medical and surgical approaches [5]. Moreover, the timing for definitive surgical interventions remains controversial [5]. Third, the myriad of etiologies makes the establishment of standardized therapeutic management difficult because diseases present several degrees of nerve recovery, and practitioners are faced with a major challenge: waiting for potential recovery versus carrying out definitive laryngeal surgery [2,4,5].

The aim of this present state-of-the art review was to investigate the current literature on the epidemiology, etiologies, diagnosis, and management of PBVCP. A focus was placed on the diagnosis approach due to its key role in the choice of therapeutic approaches and the lack of reviews dedicated to PBVCP diagnosis in the current literature. Common review recommendations were used for the selection of key papers [6,7].

## 2. Methods

The paper search was carried out with the PubMed, Scopus, and Cochrane Library databases. The databases were screened for abstracts and titles referring to the description of features of pediatric patients diagnosed with BVCP. The full texts of the selected papers were analyzed and the following information was extracted: definition, etiology, diagnosis approach, and treatment. Studies were considered if they had database abstracts, available full-texts, or titles containing the search terms. The findings of the search were reviewed for relevance. The reference lists of the included articles were examined for additional pertinent studies. The following keywords were considered: ‘bilateral’, ‘vocal cord’, ‘vocal fold’, ‘diagnosis’, ‘diagnostic’, ‘paralysis’, ‘paresis’, ‘surgery’, ‘treatment’, and ‘outcome’.

## 3. Epidemiology

PBVCP is the second most common congenital laryngeal anomaly in the pediatric population after laryngomalacia, accounting for 10% to 20% of pediatric laryngeal conditions [8]. In neonates, PBVCP represents 30% to 60% of all vocal fold dysfunctions [9]. The overall prevalence remains unclear because many cases are not detected early, especially in developing countries without sufficient otolaryngologists or head and neck surgeons [4]. However, the detection of PBVCP has improved over recent decades due to improvements in the knowledge and diagnostic approaches in pediatric otolaryngology head and neck surgery [1]. Very few papers reported incidences of PBVCP. From a population of 750,000 children, Murty et al. reported that 109 children under the age of one year required laryngoscopy because of dyspnea or stridor. Among them, 11 patients had PBVCP, which consisted of an incidence of 0.75 cases per million births per year [9]. Today, both prevalence and incidence are still poorly investigated. Prevalence depends on many points, including the hospital/center diagnostic materials, the local epidemiology of some congenital disorders associated with PBVCP, the knowledge and awareness of local pediatricians, otolaryngologists, and head and neck surgeons towards PBVCP, and, more broadly, bilateral vocal cord immobility. To be precise, in both pediatric and adult populations, there is confusion between bilateral vocal cord paralysis and posterior glottic stenosis, with some cases of posterior glottic stenosis being confounded with PBVCP, which significantly impacts the epidemiological data reports [7]. The lack of sufficient otolaryngologists or head and neck surgeons in some world regions is an additional limitation for the early detection of PBVCP. To date, there are no European or American PBVCP Registry databases collecting the cases and the related data associated with the management of these children.

## 4. Etiologies

PBVCP etiologies commonly include idiopathic, traumatic, neoplastic, infectious, congenital, and neurological conditions. To be precise, a systematic review of 320 cases reported that idiopathic, congenital, neurological, traumatic, and iatrogenic etiologies account for 42.2%, 19.7%, 16.9%, 8.6%, and 5.8% of cases, respectively [4]. Among the congenital etiologies, authors often did not identify the origin [4,5], while severe tracheal stenosis, Duane’s syndrome, or prematurity were congenital disorders commonly associated with PBVCP [4,5]. The neurological pediatric conditions include Arnold–Chiari malformations, brainstem abnormalities, Guillain–Barré syndrome, myasthenia gravis, or traumatic brain injuries [4,10,11,12]. Several traumas may lead to PBVCP, including birth anoxia, button battery ingestion, CO intoxication, and domestic or traffic accidents [4,5,13,14]. The most common iatrogenic origins concern surgeries performed on/near the laryngeal recurrent nerve path. Pediatric thyroidectomy, esophagectomy, or heart and pulmonary vessel surgeries are the most prevalent procedures associated with a risk of PBVCP [4,15,16]. The etiologies of BVCP substantially vary from children to adults. According to the most recent systematic review investigating etiologies of BVCP in adults, BVCP is iatrogenic (76.6%), idiopathic (6.9%), traumatic (3.0%), and neurological (2.5%) [7]. Among cancer etiologies, neoplasia with direct or indirect (node) contact with the laryngeal nerve may lead to PBVCP, including esophageal carcinoma, lung carcinoma, thyroid carcinoma, thymus carcinoma, laryngeal or hypopharyngeal cancer, or unspecified mediastinum neoplasia. Similarly, neurological cancer involving both vagus nerves may lead to PBVCP, but this is a very rare condition.

## 5. Clinical Presentation and Diagnosis

The PBVCP diagnosis is mainly clinical, with nasofibroscopy revealing partial or total immobility of the vocal folds. The neurological origin may be confirmed with laryngeal electromyography (LEMG) and additional examinations related to the etiology. Flexible fiberoptic endoscopy commonly reports paralyzed vocal folds in an adduction position due to abductor paralysis [1]. Primary symptoms of PBVCP include stridor, dyspnea, dysphagia, coughing, choking, apnea, or feeding difficulties. Dysphonia is not a primary symptom regarding the adduction position of the paralyzed vocal folds. However, in some cases, the vocal folds may be in an abduction position (Figure 1), which may lead to a different clinical presentation with aspiration, aphonia, and severe dysphagia. The rapid detection of PBVCP with abduction-positioned vocal folds is important due to the risk of severe pneumonia related to aspiration and penetration. In some cases, PBVCP with abduction-positioned vocal folds is related to bilateral vagus nerve paralysis, which is additionally associated with partial deinnervation of the constrictor muscles and severe dysphagia. Importantly, children with BVCP may be asymptomatic for several months, which delays the diagnosis and treatments when the child does not recover vocal fold motion [4,5]. A differential diagnosis with posterior glottic stenosis is important. During flexible fiberoptic endoscopy, the laryngeal posterior commissure and cricoarytenoid joints may appear more edematous in posterior glottic stenosis compared to PBVCP. However, it remains difficult to differentiate both conditions during the clinical examination. In direct suspension laryngoscopy, the cricoarytenoid joints are fixed in posterior glottic stenosis due to joint fibrosis. In PBVCP, the mobility of the arytenoid joints is relative or total during laryngeal palpation. Laryngologists need to document all comorbid conditions during suspension laryngoscopy, including subglottic stenosis or tracheomalacia. The differences between PBVCP and posterior glottic stenosis are difficult to objectify during suspension laryngoscopy, which, consequently, does not represent the gold standard approach for diagnosis.

The LEMG appears to be theoretically the best diagnostic tool because it may identify the nerve versus joint origin of the bilateral vocal fold immobility. However, this approach is not available in all hospitals, and pediatric otolaryngologists and head and neck surgeons are not commonly trained to perform LEMG. In contrast to adults, laryngeal pediatric EMG is commonly performed in the operating room under general anesthesia with spontaneous ventilation [17]. The electrodes are inserted into the posterior cricoarytenoid and thyroarytenoid muscles through a direct laryngoscopic approach [1,17]. The LEMG tracing of children with posterior glottic stenosis reports normal activities of the posterior cricoarytenoid muscle and thyroarytenoid muscle during inspiration and expiration, respectively. In PBVCP, the tracing is associated with electromyographic evidence of denervation [1,17]. In 2022, Aragon-Ramos et al. explored the usefulness of LEMG for distinguishing vocal cord paralysis from posterior glottic stenosis in 10 children with bilateral vocal cord immobility [17]. The authors reported pathologic laryngeal EMG abnormalities in four cases, while they did not find recurrent laryngeal nerve injury in congenital bilateral vocal fold immobility [17]. The findings of this prospective study strengthened the need to document the origin(s) of bilateral vocal fold immobility to improve clinical decision-making and patient care, especially when a congenital origin is suspected [17]. Indeed, the laryngologist may expect vocal fold motion recovery in children with idiopathic BVCP lasting for less than 1 year, while the recovery of vocal fold immobility related to posterior glottic stenosis is unexpected. However, LEMG is poorly used in daily practice because of the rarity of PBVCP, the unavailability of material in many hospitals, and the lack of physician training. Moreover, the LEMG reliability strongly depends on the experience of laryngologists and neurophysiologists [18].

In summation, the diagnosis of PBVCP may be currently based on the etiology of the child’s disorder(s), their clinical history, the flexible fiberoptic laryngoscopy findings, the assessment of comorbid tracheal or esophageal conditions, the findings reported through the suspension laryngoscopy, and the use of additional examinations for laryngeal function (LEMG) or for documenting the etiology of PBVCP. LEMG reliability is improved when used by a trained clinical team of laryngologists and/or neurophysiologists in the operating room. In practice, a significant number of cases may be diagnosed with the medical history of patients and may not require additional examinations.

## 6. Therapeutic Strategies

The management of PBVCP should address the emergency (dyspnea) and the short-to-long-term management of airway, voice, and swallowing functions. The surgical management of the airway is necessary in 60.8% of cases [4], while the child is pauci-symptomatic or asymptomatic in the remaining cases. Tracheotomy and laterofixation of the vocal fold are the two main transient surgical procedures pending a potential vocal fold motion recovery. The injection of botulinum toxin A into the cricothyroid muscle may help to restore the airway but this approach was investigated in only one cohort study including six children [19]. Definitive surgical approaches commonly include uni- or bilateral posterior transverse cordotomy, partial or total arytenoidectomy, anterior-to-posterior cricoid split, bilateral selective laryngeal reinnervation, or definitive tracheotomy [4].

### 6.1. Transient Procedures

Tracheotomy was the standard of care in the past for children with BVCP [1,9,20,21,22]. In most national otolaryngology training programs, tracheotomy can be performed by otolaryngologists and head and neck surgeons without a fellowship in pediatric laryngology, improving the child’s airway with BVCP and protecting the lungs from aspiration [1]. In addition, tracheotomy is important to avoid prolonged intubation and related sedation. Among tracheotomized children, 54% to 77% of cases may be decannulated in the weeks/months following the surgical procedure [4,9,20,21,22]. A tracheotomy is a simple procedure, but it involves a significant burden of care for families and a risk of life-threatening complications, including granulomas, cutaneous lesions, skin infections, cannula obstruction, or accidental decannulation [23,24,25].

The laterofixation of the vocal fold is a simple and reliable surgical transient procedure developed in 1970 by Langnickel and Koburg for the management of BVCP. The technique consists of the laterofixation of one vocal fold or vocal process of the arytenoid through an external or endoscopic surgical approach. This procedure is reversible, less invasive than other definitive procedures, and requires no tracheotomy. The technique was commonly used in adults with BVCP [7] and in pediatric populations [4,10,20,26,27,28,29,30]. In 1996, Triglia et al. performed external laterofixation of the vocal fold in 15 children who were successfully decannulated in 93% of cases [26]. Mathur et al. [27] and Zawadzka-Glos et al. [28] corroborated the high percentage of success of decannulation through two retrospective studies, reporting 100% decannulation rates in children with PBVCP, respectively. Overall, external or endoscopic laterofixation of the vocal fold is associated with a significant improvement in dyspnea, stridor, and weight/height percentile findings [10,29,30]. The primary complications of this approach include aspiration or neck abscess, laryngeal edema, formation of fibrin, and malposition of the suture [27,30]. The main advantages of laterofixation of the vocal fold are the lack of tracheotomy and related burden, the preservation of the laryngeal anatomy once the laterofixation is removed, and the low proportion of complications. This procedure may be particularly recommended for children with a possibility of vocal fold motion recovery. Children with a low probability of recovery may undergo definitive laryngeal procedures.

### 6.2. Definitive Procedures

The proposition of a definitive laryngeal procedure is an important decision due to the definitive modification of the child’s laryngeal anatomy and the potential consequences on voice, breathing, and swallowing. Nemry et al. reported that the timing of definitive laryngeal procedures significantly varies from one study to another [4]. A large number of surgeons perform definitive procedures in PBVCP children without waiting for potential recovery of vocal fold motion, which is a challenging issue in the management of PBVCP. In practice, definitive procedures may be proposed considering the etiology of paralysis, the timing from paralysis to surgery, and the LEMG findings. To be precise, LEMG may play an important role in decision-making if performed by a trained and experienced team. The absence of nerve response or findings associated with the potential recovery process may support definitive laryngeal surgery, while reinnervation findings may delay the need for definitive surgeries (Figure 2). As with adults, PBVCP lasting for more than one year has little chance of recovery. Some etiologies (surgical or battery traumas, cancer) are associated with a lower recovery rate compared to others, which may help to make a rapid decision.

Total arytenoidectomy was commonly carried out a few decades ago in adults and children with BVCP and consisted of the resection of one arytenoid cartilage to enlarge the posterior glottis. This approach was effective in dyspnea and breathing evaluations but led to a high percentage of aspiration and related lung infections [4]. The risk of complications associated with total arytenoidectomy may be reduced in the adult population through voice and swallowing therapies. However, it remains controversial to continue this procedure in children, who are at greater risk of severe complications and death [4]. Thus, today, total arytenoidectomy is still not recommended due to the high rate of complications.

While total arytenoidectomy has been discontinued, partial arytenoidectomy has gained in popularity over the two past decades. The first reports of partial arytenoidectomy date from 1993 (Crumley) [31] and 1996 (Remacle et al.) [32]. Partial anteromedial arytenoidectomy consists of the unilateral or bilateral ‘semi-lunar’ resection of the anteromedial part of the arytenoid (vocal process) to enlarge the respiratory glottis while preserving the phonatory glottis [7]. The use of a CO_2_ laser may improve the surgical outcomes, with less risk of hemorrhage and better healing. The surgery commonly involves both arytenoid cartilages to increase the chance of adequate postoperative outcomes (breathing). The soft tissue of the vocal folds and the cricoarytenoid joint capsule are both preserved. The procedure is not complicated but requires a certain amount of experience according to the risk of secondary posterior glottic stenosis/cricoarytenoid joint ankylosis if the posterior laser section damages the cricoarytenoid joint [4]. In this case, the edema associated with the development of posterior glottic stenosis may reduce the airway, making the partial anteromedial arytenoidectomy ineffective. In pediatric cases, the procedure outcomes were reported by Tan et al., who performed unilateral or bilateral coblation-assisted partial arytenoidectomy in 33 children [33]. The authors reported a significant subjective improvement of dyspnea in 95% of cases, but they did not provide objective measurement changes [33]. Today, there are no other pediatric studies investigating the effectiveness and voice and swallowing outcomes of partial arytenoidectomy. However, this approach reported encouraging symptom and respiratory findings in adult populations while preserving voice quality and swallowing function [7]. Importantly, the choice of partial arytenoidectomy as a therapeutic procedure needs to consider a reliable diagnosis of PBVCP. Indeed, the effectiveness of partial arytenoidectomy is still limited in posterior glottic stenosis due to edema and fibrosis tissue around the vocal process and cricoarytenoid joint [7].

Posterior transverse cordotomy is a common laryngeal procedure consisting of the transverse unilateral or bilateral section of the mucosa, ligament, and muscle of the vocal fold through cold instruments, CO_2_ laser, KTP laser, True Blue Laser, diode laser, coblation, or monopolar radiofrequency [7]. Importantly, the posterior transverse cordotomy does not involve a surgical section on the vocal process of arytenoid cartilage but may be enlarged to include the ventricular band. The depth of the tissue section may vary from one patient to another according to the severity of the immobility and related glottic closure. In most cases, the pediatric laryngologist may start with a limited unilateral posterior transverse cordotomy while preserving the contralateral vocal cord anatomy. The choice of the side may be strengthened by the LEMG findings that may indicate which laryngeal recurrent nerve is the most likely to recover and, consequently, the side to preserve [4]. As for adults, the posterior transverse cordotomy needs to preserve as much as possible the laryngeal tissues to reduce the risk of voice and swallowing complications or impairments. In pediatric populations, the effectiveness of unilateral or bilateral posterior transverse cordotomy was investigated in three studies [34,35,36]. Overall, laryngologists and head and neck surgeons reported that this approach is associated with adequate symptom relief, worsening of voice quality, and a decannulation rate ranging from 77% to 100% [34,35,36]. Postoperative granuloma is the primary complication of partial arytenoidectomy and posterior transverse cordotomy and may be medically (therapeutic abstention or anti-reflux medication) or surgically (excision) managed [33,36]. Other complications include fibrin formation, postoperative bleeding, and scarring [4].

In adult populations, some studies have been conducted to report postoperative findings of combined laryngeal surgical procedures in BVCP [7]. Some authors associated posterior transverse cordotomy with ipsilateral partial arytenoidectomy, while others used the term ‘ventriculocordectomy’ for the description of an enlarged posterior cordotomy, which consisted of the resection of the half posterior part of the ventricular band and the vocal fold [7]. The laryngeal framework surgery is an additional procedure consisting of an enlarged posterior laryngeal surgery resecting parts of arytenoid and cricoid cartilages, vocal fold tissue, and posterior commissure [7]. In pediatrics, there are no studies comparing the postoperative airway, voice, and swallowing outcomes of these procedures.

The bilateral selective laryngeal reinnervation aims to restore the posterior cricoarytenoid muscle function and the vocal fold abduction. The first case of phrenic-to-recurrent laryngeal nerve anastomosis was reported in 1983 by Crumley, who showed an improvement of glottic diameter in adults with BVCP but no active arytenoid movement [37]. There are several variants of the procedure. The extra-laryngeal neurorrhaphy between the main trunk of the recurrent laryngeal nerve and concomitant intralaryngeal transection of the adductor branches of the recurrent nerve is a variant [38]. Other variants are the phrenic-based recurrent laryngeal nerve reinnervation, the ansa cervicalis recurrent laryngeal nerve anastomosis, or the use of the upper root of the phrenic nerve and hypoglossal nerve branch [39]. The mean decannulation rate (63%) is lower compared to other techniques, ranging from 50% to 100% [20,40,41,42]. Currently, it is still difficult to determine a mean reinnervation rate due to the heterogeneity of surgical procedures and the limited number of studies including children who underwent bilateral selective laryngeal reinnervation. When classical surgical procedures fail in decannulation and symptom relief, anterior–posterior cricoid split may be proposed. This approach consists of a laryngeal framework resecting several parts of arytenoid and cricoid cartilages to enlarge the respiratory glottis. In the study by Rutter et al. [43], 73% of children were decannulated, while dyspnea and stridor significantly improved according to studies by Sedaghat et al. [44] and Windsor et al. [45]. Note that 40% of cricoid split procedures required revisions [43,44,45].

## 7. Procedure Comparison and Perspectives

To date, there are no controlled randomized studies comparing several surgical procedures with objective lung and laryngeal measurements. The studies are retrospective and include a low number of children with PBVCP, a fact that may be attributed to the disorder’s rarity. The lack of objective lung and laryngeal measurements is an important issue because evaluations of symptoms by physicians or parents, as well as laryngoscopic findings, remain subjective [46,47]. Only LEMG was used as an objective tool in studies investigating selective laryngeal reinnervation [39,41], while there are many respiratory objective measurements available in clinical practice [48,49,50,51]. Importantly, there is no international consensus on the use of various terms. The lack of consensus may increase the difficulty in comparing the outcomes of some studies where authors appear to have carried out the same procedures but used the same terms to describe different ones. The most blatant example concerns posterior transverse cordotomy [7]. Some authors used the term ‘posterior transverse cordotomy’ to describe a resection of the vocal fold, the anterior part of the vocal process of the arytenoid with or without resection of the ventricular band [7]. For this reason, in this review, terms are used according to an anatomical standpoint.

Moreover, there is no international consensus dedicated to the management of pediatric BVCP. In addition to the lack of consensus on the naming of surgical procedures, the diagnostic approach, the indication of transient or permanent procedures, and the choice of the types of procedures have never been discussed by international pediatric otolaryngological societies or groups. In practice, the establishment of international guidelines is difficult due to the heterogeneity and paucity of studies in the literature, which may limit the drawing of reliable conclusions. Future international studies could consider both the etiology and clinical profile of children in the therapeutic management. Thus, the indication of laterofixation of the vocal fold makes sense in children with a high probability of recovery, but practitioners need to consider the appropriate timing for the recovery process. In the same way, partial arytenoidectomy should be avoided in children with bilateral vocal fold immobility with an unknown origin (posterior glottic stenosis versus BVCP). The consideration of personalized medicine is another approach that could improve the care of children with this complicated and rare laryngeal condition. A proposition of personalized management of PBVCP is available in Figure 2.

## 8. Conclusions

The management of pediatric bilateral vocal fold paralysis is still not standardized. Currently, the etiology of bilateral vocal fold immobility, the in-office or suspension of laryngoscopic findings, and the potential LEMG findings are important factors in determining the origin of immobility (BVCP versus posterior glottic stenosis), as well as in deciding the most appropriate timing and types of surgical procedures. Moreover, therapeutic management may involve several transient or permanent medical or surgical approaches that are overall associated with adequate decannulation rates, improvements of symptoms, and low complication rates. The paucity of studies, the etiological heterogeneity across child populations, and the differences between teams in the management timing limit the ability to draw reliable conclusions about the superiority of one technique over the others. The consideration of the child’s characteristics and the probability of vocal fold motion recovery is important for improving the personalized management of PBVCP.

## Figures and Tables

**Figure 1 children-11-00398-f001:**
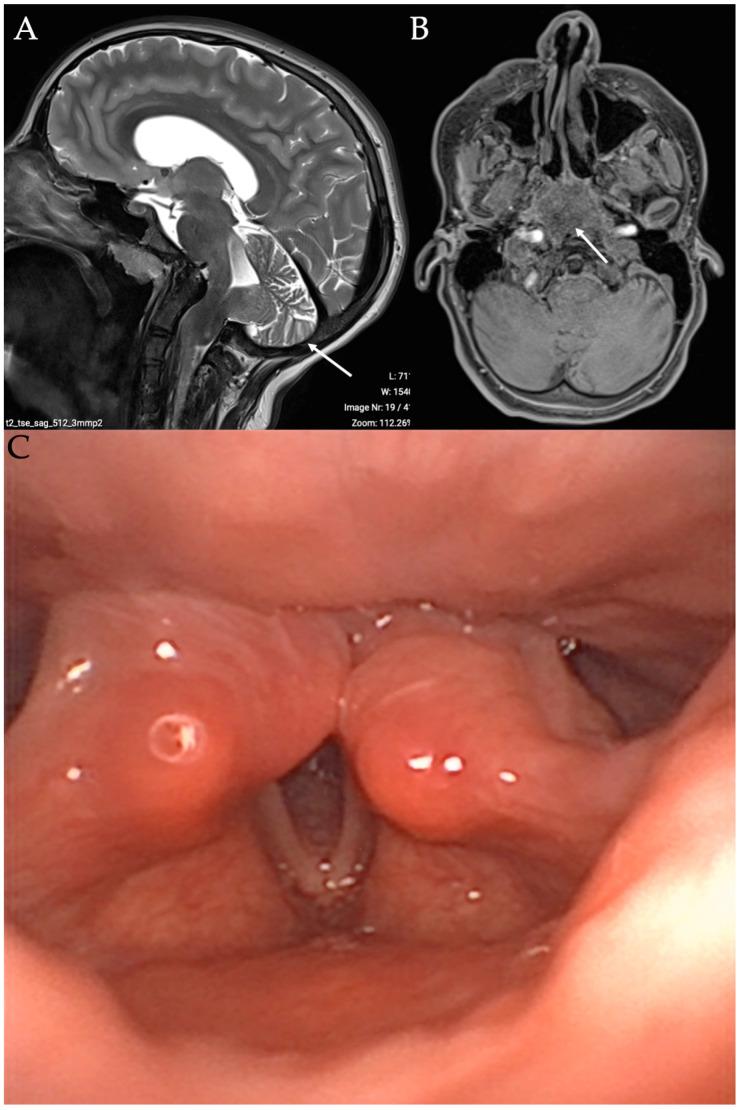
Image of bilateral vocal fold paralysis in abduction position. Image of child with bilateral vocal fold paralysis in abduction position and with Arnold–Chiari type II malformation. T1 MRI in sagittal (**A**) and axial (**B**) section reported herniation of hindbrain into low occipital or high cervical meningoencephalocele. Anatomical picture (**C**) describing anatomic relation between C1 and following cranial nerves: glossopharyngeal, vagus, and hypoglossal. Arrow is related to the Arnold-Chiari features at the imaging.

**Figure 2 children-11-00398-f002:**
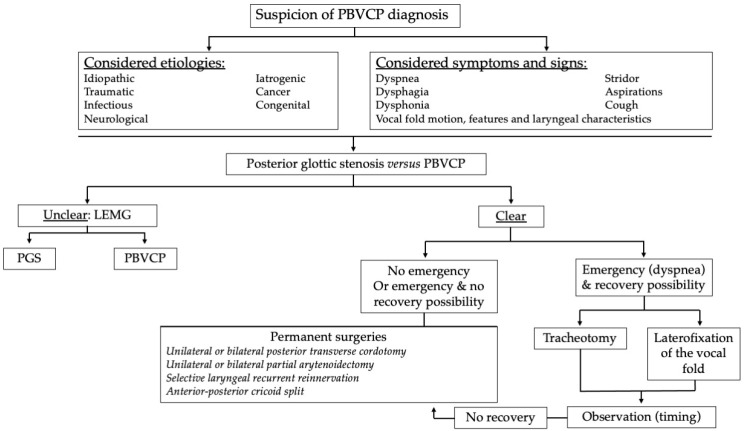
Management of pediatric bilateral vocal fold paralysis. PBVCP = pediatric bilateral vocal fold paralysis; LEMG = laryngeal electromyography; PGS = posterior glottic stenosis.

## Data Availability

No new data were created or analyzed in this study. Data sharing is not applicable to this article.

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
