# Peer review of "Management of Pediatric Bilateral Vocal Fold Paralysis: A State-of-the-Art Review of Etiologies, Diagnosis, and Treatments"

_children, 2024, doi:10.3390/children11040398_

Round 1

Reviewer 1 Report

Comments and Suggestions for Authors

The authors conducted a literature review about bilateral vocal fold paralysis in the pediatric population and its management. The article is well-documented and adds value to the existing literature. However, although your statements are supported by references, there is no information available about the methodology used to select the articles included in the manuscript. To enhance the credibility of your statements, I suggest adding a paragraph describing the literature included in your article and the primary search terms used. This will assure that the sources cited are credible and relevant to support your ideas. Additionally, it might be helpful to include statistical data wherever possible. 

Comments on the Quality of English Language

There are minor grammatical errors that need to be corrected, such as "drawing of conclusions" instead of "draw" in the conclusion section. I suggest reviewing the text and correcting any errors.

Author Response

We thank the Editor and Reviewers for the comments, we considered all of them.

First, note that a Native English Speaker has edited the paper. All modifications are in green.

Reviewer 1 

The authors conducted a literature review about bilateral vocal fold paralysis in the pediatric population and its management. The article is well-documented and adds value to the existing literature. However, although your statements are supported by references, there is no information available about the methodology used to select the articles included in the manuscript. To enhance the credibility of your statements, I suggest adding a paragraph describing the literature included in your article and the primary search terms used. This will assure that the sources cited are credible and relevant to support your ideas. Additionally, it might be helpful to include statistical data wherever possible. 

Thank you. We have added the following paragraph: “2. Methods:

The paper search was carried out with PubMED, Scopus, and Cochrane Library databases. The databases were screened for abstracts and titles referring to the description of features of pediatric patients diagnosed with BVCP. The full texts of the selected papers were analyzed and the following information were extracted: definition; etiology; diagnosis approach; treatment. Studies were considered if they had database abstracts, available full-texts or titles containing the search terms. Findings of the search were reviewed for relevance. The reference lists of the included articles were examined for additional pertinent studies. The following keywords were considered: ‘bilateral’; ‘vocal cord’; ‘vocal fold’; ‘diagnosis’; ‘diagnostic’; ‘paralysis’; ‘paresis’; ‘surgery’; ‘treatment’; ‘outcome’.

Comments on the Quality of English Language

There are minor grammatical errors that need to be corrected, such as "drawing of conclusions" instead of "draw" in the conclusion section. I suggest reviewing the text and correcting any errors.

A native speaker English has reviewed the present revised version.

Reviewer 2 Report

Comments and Suggestions for Authors

It has been a pleasure to review the proposed text about the PBVPP. The work incorporates some sound ideas, such as the need to provide a registry and more and better multicenter studies regarding treating diseases in real life using many different approaches. Almost all claim adequate results. There are some minor remarks, though.

Line 34-36: Please edit the text that is not a part of the abstract. »Keywords: keyword…

Line 54: »recurent nerve are both safe. « The author probably implies the nerves are functional while  the mobility is impaired

Line 59: »Moreoer« Moreover?

Line 81: Please check the font difference or use italics as in line 63 if necessary

Line 110: »3.0%« or better 3%

Line 199: »Granumolas« maybe granulomas?

Line 252-254: The statement seems vague, although probably mentioned in referencing the author's previous work (REF. No. 4). Breathing is the most important outcome. I propose a similar use of words.

Line 355: »Figure 2. footnotes« The term footnotes may not be needed

Author Response

Reviewer 2

It has been a pleasure to review the proposed text about the PBVPP. The work incorporates some sound ideas, such as the need to provide a registry and more and better multicenter studies regarding treating diseases in real life using many different approaches. Almost all claim adequate results. There are some minor remarks, though.

Thank you.

Line 34-36: Please edit the text that is not a part of the abstract. »Keywords: keyword…

It is corrected.

Line 54: »recurent nerve are both safe. « The author probably implies the nerves are functional while  the mobility is impaired

We have corrected: line 54, p.2: “Posterior glottic stenosis is also called cricoarytenoid joint ankylosis and the laryngeal recurrent nerve are both functional but the mobility of the cricoatytenoid joints is impaired [4].”

Line 59: »Moreoer« Moreover?

It is corrected: line 59: “Moreover, the timing for definitive surgical interventions remains controversial [5].”

Line 81: Please check the font difference or use italics as in line 63 if necessary

It is corrected. It was related to the page layout from our Word document to the template of Children.

Line 110: »3.0%« or better 3%

It is 3%, that was the most common etiologies; which do not include “others”. We have specified: line 110: “According to the most recent systematic review investigating etiologies of BVCP in adults, BVCP are iatrogenic (76.6%), idiopathic (6.9%), traumatic (3.0%), and neurological (2.5%) [7].”

Line 199: »Granumolas« maybe granulomas?

It is corrected: line 177: “The tracheotomy is a simple procedure, but it involves a significant burden of care for families and a risk of life-threatening complications, including granulomas, cutaneous lesions, skin infections, cannula obstruction or accidental decannulation [23,24].

Line 252-254: The statement seems vague, although probably mentioned in referencing the author's previous work (REF. No. 4). Breathing is the most important outcome. I propose a similar use of words.

We have corrected: “This approach was effective on dyspnea and breathing evaluations but led to a high percentage of aspiration and related lung infections [4].

Line 355: »Figure 2. footnotes« The term footnotes may not be needed

It is removed.

Reviewer 3 Report

Comments and Suggestions for Authors

The authors present a narrative review on bilateral vocal cord palsy in children to address the aetiology, diagnosis and treatment

Minor suggestions

1. Keywords need to be edited

2. The author states that PBVCP is a rare condition with an unknown prevalence; please include the reported prevalence.

3. The authors state:  The disease may affect neonates, children or infants and may be related to a myriad of neurological, malformation, iatrogenic, or idiopathic etiologies-suggest to replace the word disease

4. Most of the sentences appear lengthy and are without appropriate reference. 

5. Similar reviews are available. Please explain how this article adds to the current knowledge

Comments on the Quality of English Language

The sentences are difficult to interpret and appear lengthy.

Major grammatical, spelling and language editing is required.

Author Response

Reviewer 3

The authors present a narrative review on bilateral vocal cord palsy in children to address the aetiology, diagnosis and treatment

Minor suggestions

  1. Keywords need to be edited

Done.

  1. The author states that PBVCP is a rare condition with an unknown prevalence; please include the reported prevalence.

The prevalence is still unknow. The cases are rare but there is no study determining the prevalence. We have modified the sentence: “PBVCP is a rare condition with an undetermined prevalence [1,2].

Moreover, in the epidemiology, we have specified: line 74: “PBVCP is the second most common congenital laryngeal anomaly in the pediatric population after laryngomalacia, accounting for 10 to 20% of pediatric laryngeal conditions [8]. In neonates, PBVCP represents 30% to 60% of all vocal fold dysfunction [9]. The overall prevalence remains unclear because many cases are not detected early, especially in developing-world countries without sufficient otolaryngologists head and neck surgeons [4]. However, the detection of PBVCP has improved over the past decades due to improvements in knowledge and diagnostic approaches in pediatric otolaryngology head and neck surgery [1]. Very few papers reported incidence of PBVCP. From a population of 750,000 children, Murty et al. reported that 109 children under the age of one year required laryngoscopy because of dyspnea or stridor.”

  1. The authors state:  The disease may affect neonates, children or infants and may be related to a myriad of neurological, malformation, iatrogenic, or idiopathic etiologies-suggest to replace the word disease.

We have modified: “The disease may affect neonates, children or infants and may be related to a myriad of neurological, malformation, iatrogenic, or idiopathic diseases [1].

  1. Most of the sentences appear lengthy and are without appropriate reference. 

We have shortened the sentences, a Native English Speaker has edited the paper.

  1. Similar reviews are available. Please explain how this article adds to the current knowledge.

There is no review focusing on 1) diagnosis approach/empirical approach, and 2) the differences between posterior glottic stenosis and BVCP, which is discussed from a diagnostic standpoint and a therapeutic perspective. The most other reviews were however focused on adult populations. According to the comment of the reviewer, we have specified at the end of the introduction: “The aim of the present narrative review was to investigate the current literature about epidemiology, etiologies, diagnosis, and management of PBVCP. A focus will be placed on the diagnosis approach due to its key role in the choice of therapeutic approach, and the lack of reviews dedicated to PBVCP diagnosis in the current literature. Common review recommendations were used for the selection of key papers [6,7]..

Comments on the Quality of English Language

The sentences are difficult to interpret and appear lengthy.

Major grammatical, spelling and language editing is required.

A Native English Speaker has edited the paper.

Thanking you in advance for your attention, I remain,

Best regards,
